# ALS-LoRA: Improved Low-rank matrix compensation method for low bit quantization

## Abstract

The rapid advancement of Large Language Models (LLMs) has intensified the demand for efficient methodologies that balance model performance with hardware constraints, particularly GPU memory limitations. Quantization has emerged as a prominent technique for model compression, with QLoRA demonstrating the potential of low-rank matrices for quantization error compensation by integrating LoRA-based efficient fine-tuning. However, even LoRA fine-tuning requires substantial resources for models with tens or hundreds of billions of parameters. In this work, we explore low-rank matrix compensation for quantization errors without global LoRA fine-tuning, employing Alternating Least Squares (ALS) to better model and solve the optimization problem. We introduce a novel approach that refines low-rank matrix modeling by incorporating activation values and optimizing them directly through ALS, particularly under low-bit quantization conditions. Furthermore, we revisit the quantization interval partitioning in Round-to-Nearest (RTN) methods by introducing scaling factors that transform the discontinuous truncation function into a continuous optimization problem, thereby enhancing quantization performance through more rational interval adjustment. Extensive experimental evaluations support our theoretical contributions. Our research reveals how low-rank matrices can effectively capture the intrinsic information of large models, overcoming limitations of traditional SVD-based approaches. Comprehensive experiments across standard benchmarks consistently show that our method outperforms state-of-the-art quantization techniques, providing a principled, data-driven framework for understanding low-rank structure's role in quantization error compensation. This advancement represents a significant step toward practical LLM deployment, offering more efficient and effective model compression strategies.

## 1 Introduction

Large Language Models (LLMs) have achieved remarkable progress in recent years, demonstrating unparalleled proficiency across various language tasks, such as Natural Language Understanding (NLU) and Natural Language Generation (NLG). Despite their effectiveness, the sheer number of parameters, often in the millions or billions, poses significant challenges in terms of hardware requirements and computational resources for deployment. Consequently, there is a pressing need to compress these models effectively without compromising their performance. Among the various compression techniques, Quantization and low-rank adaptation (LoRA) are two pivotal techniques in the realm of model optimization, especially for large-scale models with billions of parameters. These techniques aim to enhance computational efficiency and reduce resource consumption while maintaining model performance.

In our research, we begin by reassessing quantization methods that primarily determine the quantization range based on extreme values, such as the Round-to-Nearest (RTN) method. Our analysis reveals that applying an appropriate scaling to these extreme values can significantly tackle the issue of outliers that deviate from the main distribution. This adjustment ensures that the quantization interval, set by these extreme values, more closely aligns with the distribution pattern of the majority of the data. Furthermore, it reduces the sensitivity of the weight matrix to variations caused by outliers. This minor enhancement proves particularly beneficial in low-bit quantization scenarios, typically involving 3 to 4 bits.

However, merely making this adjustment is insufficient, as a significant amount of information is lost during low-bit quantization. Consequently, we focus on how to efficiently compensate for this quantization error. The potential of low-rank matrices for efficient error compensation has been revealed in the efficient fine-tuning of LoRA, and there have been relevant extensions to the field of quantization. However, existing methods primarily initialize the low-rank matrix based on Singular Value Decomposition (SVD) of the quantization error of the weight matrix. For quantization problems, the impact of input values must be taken into account. Therefore, we propose employing the Alternating Least Squares (ALS) method to directly address the problem of linear layer quantization optimization. This approach enables the low-rank matrix to achieve a better compensation effect by modeling the error of the weight matrix more accurately.

On the other hand, previous work on LoRA has predominantly focused on fine-tuning the entire model. Although this approach saves a considerable amount of computational resources compared to full fine-tuning, it still poses a significant resource overhead for models with tens or even hundreds of billions of parameters. In our work, we introduce the ALS-LoRA method, which combines the theoretical compensation methods outlined earlier with further extraction of interactive information through layer-wise fine-tuning. This allows us to obtain a low-rank matrix with a good compensation effect without performing full-model LoRA, thereby greatly conserving computational resources and also aiding our further understanding of the compensation process of low-rank matrices.

In summary, our contributions are threefold:

1. **Enhanced Quantization by a Scaling Mechanism**: We enhance quantization performance by introducing a scaling mechanism that effectively handles outlier values and better aligns quantization intervals with the value distribution. This improvement significantly reduces information loss in RTN-based quantization methods.

2. **Optimized Low-rank Matrices Estimation with ALS**: We propose using the Alternating Least Squares (ALS) method for initializing low-rank matrices in LoRA. Unlike traditional Singular Value Decomposition (SVD)-based methods, ALS directly addresses the problem of linear layer quantization optimization, achieving superior compensation effects without the need for total LoRA fine-tuning.

3. **More Resource-Efficient LoRA Quantization**: We introduce the ALS-LoRA method, which combines theoretical compensation with only layerwise-finetuning. This approach significantly reduces computational overhead compared to full-model LoRA fine-tuning, making it highly efficient for large models with billions of parameters.

## 2 RELATED WORK

### 2.1 WEIGHT ONLY QUANTIZATION FOR LLMS

In the domain of weight quantization for LLMs, a substantial body of research is emerging, encompassing both Quantization-Aware Training (QAT) and Post-Training Quantization (PTQ) methodologies. QAT approaches, such as LLM-QAT (Liu et al., 2023) and EfficientQAT (Chen et al., 2025), simulate the quantization and dequantization processes during training. This enables the model to learn how to compensate for quantization-induced errors, thereby reducing quantization noise. However, as the number of parameters in large models grows, QAT becomes increasingly resource-intensive. In contrast, PTQ is far more cost-effective. Notable PTQ strategies include GPTQ (Frantar et al., 2023) and GPTAQ (Li et al., 2025), which leverage Hessian matrices, AWQ (Lin et al., 2024) and SpQR (Dettmers et al., 2023b) that address outliers with special treatment, Quarot (Ashkboos et al., 2024) and SpinQuant (Liu et al., 2025) that employ rotation matrices, and AQLM (Egiazarian et al., 2024) and VPTQ (Liu et al., 2024b) that rely on Vector Quantization. Additionally, there are hybrid methods that require only partial adjustment of training parameters, such as the low-rank matrix-based QLoRA (Dettmers et al., 2023a) method. This research landscape underscores the ongoing efforts to optimize the trade-offs between quantization accuracy and computational efficiency, particularly for large-scale models where the stakes of resource allocation are high.

## 2.2 LOW-RANK ADAPTATION(LoRA)

The fine-tuning of Large Language Models (LLMs) is often hampered by the prohibitive computational and memory costs associated with Full Fine-Tuning (FFT). Parameter-Efficient Fine-Tuning (PEFT) has emerged as a dominant paradigm to address this challenge. Among various PEFT strategies, LoRA (Hu et al., 2021) is a widely recognized parameter-efficient fine-tuning (PEFT) method in the domain of LLMs. It approximates the update of a pretrained weight matrix $\boldsymbol{W} \in \mathbb{R}^{m \times n}$ by the product of two low-rank matrices $\boldsymbol{A} \in \mathbb{R}^{m \times r}$, $\boldsymbol{B} \in \mathbb{R}^{n \times r}$, where rank $r \ll \min(m, n)$. This approach allows for significant improvements in fine-tuning performance while controlling the increase in the number of trainable parameters, and introduces no additional inference latency. The success of the original LoRA has spurred a prolific line of research into its enhancements and variants. AdaLoRA (Zhang et al., 2023) accounts for the importance of modules by introducing an adaptive budget allocation scheme, which dynamically assigns higher ranks to more critical weight matrices to improve the efficiency of the parameters. Tied-LoRA (Renduchintala et al., 2024) introduce weight tying to further reduce the complexity. DoRA (Liu et al., 2024a)decouples the pre-trained weight into magnitude and direction components, applying low-rank update solely to the direction. Notably, LoRA has been effectively combined with other compression techniques. QLoRA (Dettmers et al., 2023a)integrates 4-bit double quantization with LoRA. The performance of these methods is often sensitive to the choice of the rank hyperparameter, which requires tedious manual tuning. Moreover, the theoretical underpinnings of why the low-rank assumption holds so effectively across diverse tasks are not yet fully understood.

## 2.3 ALTERNATING LEAST SQUARES (ALS)

The Alternating Least Squares (ALS) algorithm, known for its strong optimization capabilities, is extensively utilized in recommendation systems and natural language processing. The foundational ALS algorithm has been continuously improved through a series of research efforts focused on increasing its robustness, scalability, and applicability. For example, in recommendation systems, ALS is used to handle large-scale user-item interaction matrices, employing low-rank factorization to capture the underlying relationships between users and items (Clark et al., 2018). In natural language processing, ALS is also applied to topic models, such as Latent Dirichlet Allocation (LDA), to discover latent topics within a collection of documents (Blei et al., 2003). The use of ALS for low-rank matrix recovery has been introduced (Haldar & Hernando, 2009). However, its potential for direct quantization error minimization has not been explored in the area of Large Language Model (LLM) quantization. Our work bridges this gap by reintroducing ALS to more effectively address the low-rank quantization compensation problem.

## 3 METHOD

In this section, we delve deeper into our ALS-LoRA quantization method, which primarily comprises three key components: the optimized RTN quantization foundation, the computation of the compensatory low-rank matrix, and the synergistic enhancement of the low-rank matrix's compensatory capabilities through layerwise-finetuning.

### 3.1 LOW RANK MATRIX OPTIMIZATION PROBLEM

We begin by modeling the optimization problem for the quantization error matrix of a linear layer. Given a pre-trained weight matrix $\boldsymbol{W} \in \mathbb{R}^{m \times n}$, a $N$-bit quantized weight matrix $\boldsymbol{Q} \in \mathbb{R}^{m \times n}$, and low-rank matrices $\boldsymbol{A} \in \mathbb{R}^{m \times r}$ and $\boldsymbol{B} \in \mathbb{R}^{n \times r}$ with rank $r \ll \min(m, n)$, the objective can be formulated as minimizing the quantization error through the following optimization problem:

$$\min_{\widehat{\boldsymbol{W}}} \|\boldsymbol{W} - \widehat{\boldsymbol{W}}\|_2^2 = \min_{\boldsymbol{A}, \boldsymbol{B}} \|\boldsymbol{W}\boldsymbol{X} - (\boldsymbol{Q} + \boldsymbol{A}\boldsymbol{B}^T)\boldsymbol{X}\|_2^2 = \min_{\boldsymbol{A}, \boldsymbol{B}} \|\Delta\boldsymbol{W}\boldsymbol{X} - \boldsymbol{A}\boldsymbol{B}^T\boldsymbol{X}\|_2^2 \quad (1)$$

where $\Delta\boldsymbol{W} = \boldsymbol{W} - \boldsymbol{Q}$ represents the quantization error. The main goal of Equation(1) is to minimize this error by optimizing over the low-rank matrices $\boldsymbol{A}$ and $\boldsymbol{B}$. This optimization process aims to maximize the compensation for the quantization error.

## 3.2 SCALING MECHANISM

For the N-bit Round to Nearest quantization, the quantization and dequantization processes can be described as follows:

$$Q_{z,s}(\boldsymbol{W}) = \text{round}\left(\frac{\boldsymbol{W}}{s} + z\right) = \boldsymbol{W_q} \tag{2}$$

$$Q_{z,s}^{-1}(\boldsymbol{W_q}) = s(\boldsymbol{W_q} - z) \tag{3}$$

where the quantization scale is $s = (\boldsymbol{W}_{\max} - \boldsymbol{W}_{\min})/(2^N - 1)$, and the zero-point is $z = -\boldsymbol{W}_{\min}/s$.

According to the principle of RTN, the quantization scale $s$ is primarily determined by $\boldsymbol{W}_{\max}$ and $\boldsymbol{W}_{\min}$, which means it is mainly influenced by the extreme values at both ends of the weight matrix distribution.However, the majority of parameters in the weight matrix are concentrated in a narrower range, with only a few outliers that significantly affect the quantization interval division, leading to greater precision loss for the central values. Therefore, we propose introducing a scaling factor $\eta$ to adjust $\boldsymbol{W}_{\max}$ and $\boldsymbol{W}_{\min}$. enabling the calculated interval to better approximate the distribution of the majority of parameters. Yet, the weight matrix is more sensitive to these outliers than to other values in the middle of the distribution. Thus, an excessively small $\eta$ would lead to precision degradation rather than improvement. Consequently, $\eta$ needs to be chosen as an appropriate value within the range $(0, 1)$, which can be easily implemented through a step-by-step search method.

Although quantile-based truncation methods exist for quantization interval processing, they primarily rely on discontinuous truncation functions that can cause significant abrupt changes around outlier values, adversely affecting overall error. In contrast, our introduced scaling factor enables continuous adjustment of boundary values, achieving a better balance between errors in the central distribution and those from outliers. Furthermore, as demonstrated in Section4.1, this adjustment approach proves effective not only in RTN quantization but also in other quantization methods that require boundary interval configuration.

Subsequently, we can further refine the quantization intervals by incorporating the low-rank compensation matrices $\boldsymbol{A}$ and $\boldsymbol{B}$. Similar to LoftQ (Li et al., 2023), we initialize matrices $\boldsymbol{A}$ and $\boldsymbol{B}$ using the first r singular vectors obtained from SVD. We then refine $\boldsymbol{A}$, $\boldsymbol{B}$, and the RTN quantization through Alternating Optimization to achieve improved performance. Specifically, we perform SVD on the quantization error to acquire the top r singular vectors, which serve as the initial values for $\boldsymbol{A}$ and $\boldsymbol{B}$. Concurrently, we update the quantization scheme to accommodate the updates to $\boldsymbol{A}$ and $\boldsymbol{B}$. This process is detailed in Algorithm 1.

---

**Algorithm 1** Adjust Quantization Scale and Init $\boldsymbol{A}$ , $\boldsymbol{B}$

---

1: **Require:** Full precision weight matrix $\boldsymbol{W}$, quantization bit $N$, scaling factor $\eta$, rank $r$, number of iterations $T$.
2: Initialize $\boldsymbol{A}$, $\boldsymbol{B} \leftarrow 0, 0$
3: **for** epoch $\tau$ in $T$ **do**
4:      Calculate $\boldsymbol{W}_{\max}$ $\boldsymbol{W}_{\min} \leftarrow \boldsymbol{W} - \boldsymbol{A}\boldsymbol{B}^T$
5:      $\boldsymbol{W}_{\max} = \eta \times \boldsymbol{W}_{\max}$ , $\boldsymbol{W}_{\min} = \eta \times \boldsymbol{W}_{\min}$
6:      Quantization scale $s$, zero-point $z \leftarrow \boldsymbol{W}_{\max}, \boldsymbol{W}_{\min}$
7:      $\boldsymbol{W_q} \leftarrow s, z$
8:      Calculate dequantization weight matrix and error $\boldsymbol{Q} \leftarrow \boldsymbol{W_q}, s, z, \Delta\boldsymbol{W} \leftarrow \boldsymbol{W} - \boldsymbol{Q}$
9:      $\boldsymbol{A}$, $\boldsymbol{B} \leftarrow SVD(\Delta\boldsymbol{W}), r$
10: **end for**
11: **Return** $N$-bit quantization weight matix $\boldsymbol{W_q}$, Init Low-rank matrices $\boldsymbol{A}$, $\boldsymbol{B}$

---

## 3.3 ALTERNATING LEAST SQUARES (ALS) OPTIMIZATION

Unlike the previous method that used SVD decomposition to fit $\Delta\boldsymbol{W}$ with matrices $\boldsymbol{A}$ and $\boldsymbol{B}$, we employ Alternating Least Squares (ALS) to directly solve Equation(1). To enhance the robustness of the fit, we added a regularization term to Equation(1) and then formulated the optimization problem

as follows:

$$\ell = \|\Delta \boldsymbol{W} \boldsymbol{X} - \boldsymbol{A} \boldsymbol{B}^T \boldsymbol{X}\|_2^2 + \lambda(\|\boldsymbol{A}\|_2^2 + \|\boldsymbol{B}^T \boldsymbol{X}\|_2^2)$$
$$= tr[\Delta \boldsymbol{W} \boldsymbol{X} \boldsymbol{X}^T \Delta \boldsymbol{W}^T - \Delta \boldsymbol{W} \boldsymbol{X} \boldsymbol{X}^T \boldsymbol{B} \boldsymbol{A}^T - \boldsymbol{A} \boldsymbol{B}^T \boldsymbol{X} \boldsymbol{X}^T \Delta \boldsymbol{W}^T +$$
$$\boldsymbol{A} \boldsymbol{B}^T \boldsymbol{X} \boldsymbol{X}^T \boldsymbol{B} \boldsymbol{A}^T + \lambda(\boldsymbol{A} \boldsymbol{A}^T + \boldsymbol{B}^T \boldsymbol{X} \boldsymbol{X}^T \boldsymbol{B})] \tag{4}$$

$$\frac{\partial \ell}{\partial \boldsymbol{A}} = -\Delta \boldsymbol{W} \boldsymbol{H} \boldsymbol{B} + \boldsymbol{A} \boldsymbol{B}^T \boldsymbol{H} \boldsymbol{B} + 2\lambda \boldsymbol{A} \tag{5}$$

$$\frac{\partial \ell}{\partial \boldsymbol{B}} = -\boldsymbol{H} \Delta \boldsymbol{W}^T \boldsymbol{A} + \boldsymbol{H} \boldsymbol{B} \boldsymbol{A}^T \boldsymbol{A} + \lambda \boldsymbol{H} \boldsymbol{B} \tag{6}$$

where $\boldsymbol{H} = 2\boldsymbol{X} \boldsymbol{X}^T$ is mathematically referred to as the Hessian matrix, which can be estimated using calibration data. The parameter $\lambda$ denotes the regularization coefficient, typically assigned a small value (e.g., $1 \times 10^{-5}$).

The ALS method is fundamentally about reformulating the original optimization problem into two interdependent convex quadratic programming subproblems. This is achieved by iteratively fixing one matrix while optimizing the other. These alternating steps continue iteratively until $\boldsymbol{A}$ and $\boldsymbol{B}$ reach stable values, which represent the final solution to the optimization problem. The iterative nature of this method capitalizes on the convexity of the subproblems, ensuring that each iteration results in an improvement in the objective function, ultimately leading to a solution that minimizes the overall error.To expedite convergence, we utilize the values obtained in Algorithm 1 as the initialization for the ALS algorithm.

To optimize matrix $\boldsymbol{A}$ while holding matrix $\boldsymbol{B}$ constant, we set Equation (5) to zero:

$$\boldsymbol{A} = \Delta \boldsymbol{W} \boldsymbol{H} \boldsymbol{B}(\boldsymbol{B}^T \boldsymbol{H} \boldsymbol{B} + 2\lambda \boldsymbol{I})^{-1} \tag{7}$$

Similarly, to optimize matrix $\boldsymbol{B}$ while keeping matrix $\boldsymbol{A}$ fixed, we set Equation (6) to zero:

$$\boldsymbol{B} = \Delta \boldsymbol{W}^T \boldsymbol{A}(\boldsymbol{A}^T \boldsymbol{A} + \lambda \boldsymbol{I})^{-1} \tag{8}$$

Algorithm2 provides a comprehensive summary of the procedure outlined in this subsection.

---

**Algorithm 2** ALS for Low-Rank Matrix Estimation of $\boldsymbol{A}$ and $\boldsymbol{B}$

---

1: **Require:** Quantization error $\Delta \boldsymbol{W}$, Hessian matrix $\boldsymbol{H}$, input value $\boldsymbol{X}$, scaling factor $\lambda$, number of iterations $T$.
2: Initialize $\boldsymbol{A}$, $\boldsymbol{B} \leftarrow$ **Algorithm 1**
3: Initialize best error $\epsilon_0 \leftarrow 0$
4: **for** epoch $\tau$ in $T$ **do**
5:    Fix $\boldsymbol{B}$ and update $\boldsymbol{A}$: $\boldsymbol{A} \leftarrow \Delta \boldsymbol{W} \boldsymbol{H} \boldsymbol{B}(\boldsymbol{B}^T \boldsymbol{H} \boldsymbol{B} + 2\lambda \boldsymbol{I})^{-1}$
6:    Fix $\boldsymbol{A}$ and update $\boldsymbol{B}$: $\boldsymbol{B} \leftarrow \Delta \boldsymbol{W}^T \boldsymbol{A}(\boldsymbol{A}^T \boldsymbol{A} + \lambda \boldsymbol{I})^{-1}$
7:    Calculate current mean squared error: $\epsilon \leftarrow \frac{1}{n} \sum_{i=1}^{n} (\Delta \boldsymbol{W} \boldsymbol{X} - \boldsymbol{A} \boldsymbol{B}^T \boldsymbol{X})_i^2$
8:    **if** $\epsilon < \epsilon_0$ and $\epsilon_0 > 0$ **then**
9:      Update best error: $\epsilon_0 \leftarrow \epsilon$
10:    **else if** $\epsilon \geq \epsilon_0$ **then**
11:      Early break.
12:    **end if**
13: **end for**
14: **Return** Low-rank matrices $\boldsymbol{A}$, $\boldsymbol{B}$

---

### 3.4 LAYERWISE-FINETUNING

In Section 3.3, we computed the low-rank compensation matrices $\boldsymbol{A}$ and $\boldsymbol{B}$ for individual linear modules using the Alternating Least Squares (ALS) method. To further enhance the ability of

these low-rank matrices to capture relationships between different linear modules within the same transformer layer, while also balancing reduced resource consumption and maintained performance, we can implement additional layer-wise fine-tuning. The experiments in Section 4.3.2 demonstrate that this serves as an effective additional improvement to further enhance the compensation effect of the low-rank matrices.

# 4 EXPERIMENTS

**Models and Datasets:** Our evaluation framework employs three large language models: Llama3.1-8B (AI, 2023), Mistral-7B (Jiang et al., 2023), and Llama-30B (Touvron et al., 2023). Model performance is assessed using two key metrics: perplexity (PPL) and accuracy (Acc). For perplexity evaluation, we utilize the Wikitext2 (Merity et al., 2016) and C4 (Raffel et al., 2019) datasets, while accuracy is measured on the ARC-Challenge (ARC-c) and ARC-Easy (ARC-e) (Clark et al., 2018) benchmarks.

## 4.1 EFFECT OF SCALING FACTOR $\eta$

Table 1: Effect of scaling factor $\eta$.

| Bits | $\eta$ | Wiki2($\downarrow$) | Bits | $\eta$ | Wiki2(($\downarrow$) | Bits | quantile | Wiki2($\downarrow$) |
|---|---|---|---|---|---|---|---|---|
| FP16/16 | - | 5.65 | FP16/16 | - | 5.65 | FP16/16 | - | 5.65 |
| RTN/4 | 1.0 | 7.62 | RTN/3 | 1.0 | 750.69 | RTN(cmp)/3 | 1.0 | 750.69 |
| | 0.9 | 6.95 | | 0.9 | 178.15 | | 0.99998 | 534.88 |
| | **0.8** | **6.73** | | 0.8 | 89.12 | | 0.99995 | 407.73 |
| | 0.7 | 6.73 | | 0.7 | 32.08 | | 0.99990 | 222.49 |
| | 0.6 | 7.03 | | **0.6** | **13.93** | | 0.99985 | 177.66 |
| | 0.5 | 8.70 | | 0.5 | 19.82 | | **0.99982** | **171.47** |
| | 0.4 | 173.72 | | 0.4 | 454.38 | | 0.99980 | 6159.86 |
| Had/4 | 1.0 | 7.03 | Had/3 | 1.0 | 1089.94 | Had(cmp)/3 | 1.0 | 1089.94 |
| | 0.9 | 6.69 | | 0.9 | 63.57 | | 0.9998 | 104.72 |
| | 0.8 | 6.49 | | 0.8 | 30.21 | | 0.9996 | 52.84 |
| | **0.7** | **6.34** | | 0.7 | 13.94 | | 0.9995 | 43.57 |
| | 0.6 | 6.49 | | **0.6** | **10.36** | | 0.9994 | 35.96 |
| | 0.5 | 7.32 | | 0.5 | 11.16 | | **0.9992** | **29.53** |
| | 0.4 | 75.17 | | 0.4 | 461.03 | | 0.9991 | 33.47 |
| PoT/4 | 1.0 | 7.68 | PoT/3 | 1.0 | 1240.03 | PoT(cmp)/3 | 1.0 | 1240.03 |
| | 0.9 | 7.56 | | 0.9 | 164.64 | | 0.99995 | 265.94 |
| | 0.8 | 7.53 | | 0.8 | 64.33 | | 0.99990 | 141.62 |
| | **0.7** | **6.94** | | 0.7 | 18.43 | | 0.99988 | 88.94 |
| | 0.6 | 7.78 | | 0.6 | 13.36 | | **0.99985** | **56.99** |
| | 0.5 | 7.44 | | **0.5** | **12.28** | | 0.99982 | 109.97 |
| | 0.4 | 7.40 | | 0.4 | 130.88 | | 0.99980 | 593.90 |

In this section, we present our experimental findings, which were primarily conducted using the Llama3.1-8B model. The performance of the quantized model was evaluated using Perplexity (PPL) as the test metric on the Wikitext2 dataset. The objective of these experiments is to illustrate that selecting an appropriate scaling factor, denoted as $\eta$, can markedly improve the efficacy of the quantization process. Furthermore, in addition to the basic Round to Nearest (RTN) method, we also assessed the impact of applying the scaling factor $\eta$ to other techniques. These included an enhanced version of RTN that employs Hadamard rotation matrix transformations to mitigate outliers (Tseng et al., 2024), as well as a non-uniform quantization method known as PoT (Miyashita et al., 2016), which is based on powers of two.

The results are presented in Table 1, where the first three columns from the left represent 4-bit quantization with scaling factor $\eta$, the middle three columns represent 3-bit quantization with scaling factor $\eta$, and the last three columns represent the traditional 3-bit quantization method using direct tail truncation based on quantiles, which is compared with the middle three columns. For each scenario,

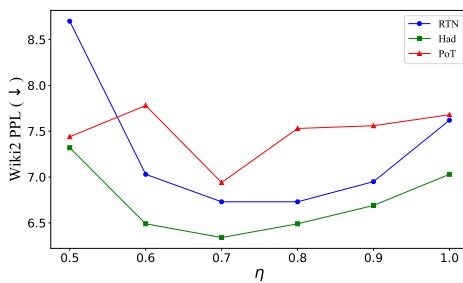 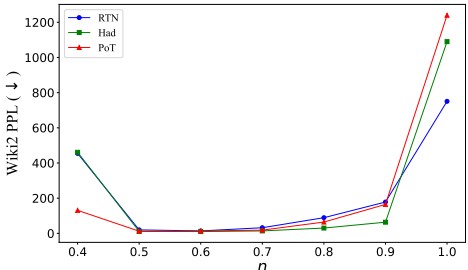

Figure 1: 4-bit quantization PPL changes with different $\eta$.

Figure 2: 3-bit quantization PPL changes with different $\eta$.

we tested the RTN method, the Hadamard-improved RTN, and the non-uniform Power-of-Two (PoT) quantization. Figure 1 and Figure 2 illustrate the variations in PPL as the scaling factor $\eta$ is adjusted under 4-bit and 3-bit quantization scenarios, respectively. The results indicate that after applying an appropriate scaling factor $\eta$, all three methods—4-bit and 3-bit quantizations—showed significant improvements over the original quantization results, with particularly notable enhancements for 3-bit quantization, where PPL could be reduced to a few tenths or even a hundredth of the original value. Additionally, the results demonstrate that across a relatively wide range of $\eta$ values, there is consistently good performance, which is more convenient for practical engineering applications and parameter tuning. When comparing the methods using $\eta$ scaling with those using quantile truncation for 3-bit quantization, it is evident that quantile truncation is highly sensitive to the setting of the quantile and does not effectively preserve information about outliers, resulting in a noticeably inferior performance compared to the $\eta$ scaling method.

Overall, the experimental results suggest that adjusting the scaling factor method does not require additional storage and has low computational overhead, making it a promising approach for improving low-bit quantization in interval-based segmentation.

### 4.2 QUANTIZATION PERFORMANCE

During the evaluation of our ALS-LoRA quantization approach, we utilized the Round to Nearest (RTN) quantization model as our primary baseline to assess the low-bit quantization performance on the Llama3.1-8B, Mistral-7B, and Llama-30B models. This included evaluating the compensation effects at 4-bit, 3-bit, and 2-bit levels. In our experimental setup, for 4-bit and 3-bit RTN, we employed per-channel quantization, while for 2-bit, we used per-group quantization with group sizes of 32 and 64. For ALS-LoRA, the scaling factor $\eta$ was set to 0.8 for 4-bit quantization on both Llama3.1-8B and Mistral-7B, and to 0.6 for 3-bit and 2-bit quantization. For Llama-30B, $\eta$ was set to 0.9 for 4-bit quantization and 0.8 for 3-bit quantization. The rank of low-rank matrices was 32. The calibration dataset required for ALS-LoRA consisted of 128 samples randomly selected from the Wikitext2 dataset, and the layer-wise fine-tuning was performed for 3 epochs. In terms of testing, we primarily measured two metrics: Perplexity (PPL) and Accuracy. PPL was assessed using the Wikitext2 and C4 datasets, while Accuracy was evaluated using the question-answering datasets ARC-Challenge (ARC-c) and ARC-Easy (ARC-e). The results were recorded in Table 2.

The findings clearly demonstrate that ALS-LoRA has outperformed the baseline significantly. In the case of 4-bit quantization, ALS-LoRA's PPL on Wikitext2 is within 0.5 of that achieved by FP16, and its accuracy is within 5%, indicating an effective compensation. For 3-bit quantization, ALS-LoRA offers even greater compensation compared to RTN, potentially reducing the PPL to a fraction or even a tenth of its original value. The table results show that ALS-LoRA exhibits strong compensation capabilities for both 4-bit, 3-bit, and 2-bit quantization. For instance, when quantizing the Llama 3.1-8B model at 3 bits, ALS-LoRA reduces the PPL on Wikitext2 from 750.7 to 7.1, a decrease to a little over one percent of the original value. Similarly, for 2-bit quantization, the PPL on Wikitext2 is dramatically reduced from 4346.10 to 10, a reduction of several orders of magnitude. These results

underscore the robustness and effectiveness of ALS-LoRA in enhancing the performance of quantized models, particularly in low-bit scenarios.

In summary, the ALS-LoRA method has shown good compensation effects across different quantization bit numbers and model sizes. In low-bit quantization scenarios, especially for large language models, it provides an effective strategy for compensating quantization errors, allowing for a reduction in model size while maintaining performance as much as possible. This is particularly significant for deploying large language models in resource-constrained environments.

Table 2: Performance comparing.

| Model | Bits | Method | PPL(↓) | | Accuracy(↑) | |
| | | | Wiki2 | C4 | Arc-c | Arc-e |
|---|---|---|---|---|---|---|
| | 16 | FP16 | 5.65 | 9.40 | 79.6% | 91.3% |
| | 4 | RTN | 7.62 | 13.06 | 68.4% | 82.5% |
| | 4 | RTN($\eta = 0.8$) | 6.73 | 11.51 | 69.5% | 85.9% |
| | 4 | ALS-LoRA | 6.09 | 10.49 | 75.9% | 89.9% |
| Llama3.1-8B | 3 | RTN | 750.7 | 805.3 | 3.0% | 4.0% |
| | 3 | RTN($\eta = 0.6$) | 13.93 | 23.37 | 35.0% | 50.1% |
| | 3 | ALS-LoRA | 7.10 | 12.72 | 55.2% | 72.5% |
| | 2 | RTN(g32) | 4346.10 | 780.86 | 7.3% | 7.7% |
| | 2 | ALS-LoRA(g32) | 10.60 | 19.85 | 34.2% | 41.2% |
| | 16 | FP16 | 6.81 | 7.91 | 73.5% | 85.0% |
| | 4 | RTN | 7.78 | 8.90 | 45.2% | 68.2% |
| | 4 | RTN($\eta = 0.8$) | 7.67 | 8.74 | 58.1% | 78.5% |
| | 4 | ALS-LoRA | 6.97 | 8.13 | 71.2% | 84.3% |
| Mistral-7B | 3 | RTN | 78.25 | 53.53 | 17.7% | 19.8% |
| | 3 | RTN($\eta = 0.6$) | 21.60 | 37.03 | 11.3% | 18.6% |
| | 3 | ALS-LoRA | 7.50 | 8.85 | 58.3% | 76.2% |
| | 2 | RTN(g32) | 25.03 | 21.20 | 11.8% | 11.5% |
| | 2 | ALS-LoRA(g32) | 11.35 | 12.55 | 38.8% | 39.3% |
| | 2 | RTN(g64) | 173.63 | 78.79 | 7.0 % | 7.7% |
| | 2 | ALS-LoRA(g64) | 12.00 | 13.70 | 32.0% | 39.3% |
| | 16 | FP16 | 5.13 | 6.20 | 64.2% | 78.9% |
| | 4 | RTN | 5.68 | 6.60 | 59.6% | 71.9% |
| | 4 | RTN($\eta = 0.9$) | 5.56 | 6.50 | 63.7% | 76.1% |
| Llama-30B | 4 | ALS-LoRA | 5.32 | 6.33 | 64.0% | 77.2% |
| | 3 | RTN | 24.85 | 25.85 | 31.1% | 42.3% |
| | 3 | RTN($\eta = 0.8$) | 9.37 | 9.42 | 43.3% | 54.2% |
| | 3 | ALS-LoRA | 6.21 | 7.22 | 55.0% | 64.1% |

## 4.3 ABLATION STUDIES

### 4.3.1 INFLUENCE OF RANK

In this section, we evaluate the low-rank matrix compensation effectiveness of ALS-LoRA under different rank configurations. Based on the Llama3.1-8B model, we experiment with various rank settings for both 4-bit and 3-bit quantization. The performance is assessed using perplexity (PPL) on Wikitext2 and C4 datasets, with results summarized in Table 3. Experimental results demonstrate that as the rank increases, the PPL correspondingly decreases, indicating that higher ranks lead to better compensation effects, albeit at the cost of increased additional storage overhead.

Table 3: Results under different rank

| Bits | Rank | PPL(↓) | | Bits | Rank | PPL(↓) | |
|------|------|--------|------|------|------|--------|------|
| | | Wiki2 | C4 | | | Wiki2 | C4 |
| 16 | - | 5.65 | 9.40 | 16 | - | 5.65 | 9.40 |
| 4 | 64 | 6.01 | 10.42 | 3 | 64 | 6.82 | 12.34 |
| 4 | 32 | 6.09 | 10.49 | 3 | 32 | 7.10 | 12.72 |
| 4 | 16 | 6.15 | 10.52 | 3 | 16 | 7.34 | 12.73 |
| 4 | 8 | 6.21 | 10.62 | 3 | 8 | 7.58 | 12.94 |

### 4.3.2 INFLUENCE OF LAYERWISE-FINETUNE

To assess the benefits of layer-wise fine-tuning in our approach, we conducted the ablation study presented in Table 4 Based on the Llama3.1-8B model, we performed quantization experiments to compare the model's performance on the test set before and after incorporating layer-wise fine-tuning. The results indicate that even without enabling layer-wise fine-tuning, the address matrix computed by ALS-LoRA achieves a reasonably good compensation effect. Layerwise-finetuning serves as an additional enhancement, further improving the compensatory capabilities of the low-rank matrix. Moreover, compared to full-model LoRA fine-tuning, layer-wise fine-tuning consumes significantly fewer GPU resources.

Table 4: Influence of layerwise-finetune

| Method | Bits | Rank | PPL(↓) | | Accuracy(↑) | |
|--------|------|------|--------|------|-------------|------|
| | | | Wiki2 | C4 | Arc-c | Arc-e |
| FP16 | - | - | 5.65 | 9.40 | 79.6% | 91.3% |
| RTN | 4 | - | 7.62 | 13.06 | 68.4% | 82.5% |
| ALS-LoRA(no fine) | 4 | 16 | 6.28 | 10.76 | 74.1% | 88.5% |
| ALS-LoRA | 4 | 16 | 6.03 | 10.59 | 75.9% | 89.0% |
| RTN | 3 | - | 750.7 | 805.3 | 3.0% | 4.0% |
| ALS-LoRA(no fine) | 3 | 16 | 7.89 | 13.99 | 54.6% | 71.0% |
| ALS-LoRA | 3 | 16 | 7.34 | 13.08 | 60.0% | 79.1% |

## 5 CONCLUSION

This work addresses the critical challenge of deploying large language models under hardware constraints by introducing an optimized quantization framework. The key innovation lies in developing a low-rank matrix compensation method that bypasses the resource-intensive LoRA fine-tuning through Alternating Least Squares (ALS) optimization. Our approach enhances quantization along three dimensions: (1) it incorporates activation values into low-rank modeling via ALS for better error compensation, particularly in low-bit settings; (2) it introduces scaling factors to transform discontinuous truncation into continuous optimization, improving interval adjustment in interval-based quantization; (3) it introduces layer-wise fine-tuning to further enhance the compensation effect of the low-rank matrix. Extensive experiments demonstrate superior performance over existing methods, providing both theoretical insights and practical advantages for efficient LLM deployment.

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
