# OpenReview forum: "ALS-LoRA: Improved Low-Rank Matrix Compesation Method for Low Bit Quantization"
_ICLR.cc/2026/Conference — Submitted to ICLR 2026_

### Official Review · Reviewer_waPS · 2025-10-29

**Soundness:** 2
**Presentation:** 2
**Contribution:** 2
**Rating:** 2
**Confidence:** 5

**Summary:**

This paper presents ALS-LoRA, a method for low-rank error compensation in the context of post-training quantization (PTQ) for LLMs. The method combines three main ideas: (1) a scalar hyperparameter η to adjust the quantization range in Round-to-Nearest (RTN) quantization; (2) using Alternating Least Squares (ALS) to optimize the low-rank factors by incorporating activation data; and (3) an optional layer-wise fine-tuning step. The paper reports significant perplexity and accuracy improvements for this method when compared against a baseline RTN quantizer.

**Strengths:**

1. Problem Formulation: The paper formulates the low-rank compensation problem (Eq. 1) to include the activation matrix X, which is a standard step in output-aware optimization.
2. Reported Gains Over Baseline: The experiments show that applying the proposed techniques leads to large performance gains over a basic RTN quantizer, especially in 2-bit and 3-bit settings.

**Weaknesses:**

1. Fatal Flaw: Use of a Strawman Baseline and Complete Omission of Relevant Literature: This is an unacceptable and disqualifying weakness. The paper's entire empirical validation rests on comparing "ALS-LoRA" to the Round-to-Nearest (RTN) method, which is the most naive form of PTQ and has not been a competitive baseline for years.
    ○ Ignoring Established Standards: The paper fails to compare against even the most basic, established standards from 2023, such as GPTQ and AWQ. Any serious work in PTQ must, at a minimum, demonstrate superiority over these widely adopted methods.
    ○ Ignoring the Entire Modern SOTA: More damningly, the paper completely disregards the significant progress made in 2024 and 2025. There is no mention, let alone comparison, to a vast body of recent, highly relevant work from top-tier conferences, such as DuQuant[1] (NeurIPS 2024), OSTQuant[2] (ICLR 2025), AMLQ[3] (COLM 2024), or other data-aware methods like GPTAQ[4] (ICML 2025) and GuidedQuant[5] (ICML 2025).The impressive-looking perplexity reductions in Table 2 are misleading and scientifically invalid, as they only prove the unsurprising fact that any reasonable optimization is better than none.
2. Gross Overstatement of Novelty: The paper frames its "Scaling Mechanism" (η) as a key contribution. This is a gross overstatement. Adjusting the clipping range for quantization is a rudimentary form of hyperparameter tuning, not a novel method. Presenting this as a core contribution is misleading and highlights a lack of substance in the work's actual novelty.
3. Contradictory Motivation and Opaque Cost Analysis: The paper motivates its approach by citing the high cost of LoRA fine-tuning, yet paradoxically introduces its own "layerwise-finetuning" step without providing any cost analysis. This is a direct contradiction. The paper provides no details on the time, memory, or data needed for this step, leaving the reader unable to assess the method's true resource efficiency.
4. Conflated and Uninterpretable Ablations: The paper's structure makes it impossible to disentangle the effects of its three components. The main results table presents "ALS-LoRA" as a monolithic block. It is unclear whether the reported gains come from the simple η-tuning, the ALS optimization, the layerwise-finetuning, or some combination. This lack of rigorous ablation makes the source of any improvement unknowable.

[1] Lin, Haokun, et al. "DuQuant: Distributing Outliers via Dual Transformation Makes Stronger Quantized LLMs." The Thirty-eighth Annual Conference on Neural Information Processing Systems.
[2] Hu, Xing, et al. "OSTQuant: Refining Large Language Model Quantization with Orthogonal and Scaling Transformations for Better Distribution Fitting." The Thirteenth International Conference on Learning Representations.
[3] Li, Ou, et al. "Adaptive Quantization Error Reconstruction for LLMs with Mixed Precision." First Conference on Language Modeling.
[4] Li, Yuhang, et al. "GPTAQ: Efficient Finetuning-Free Quantization for Asymmetric Calibration." Forty-second International Conference on Machine Learning.
[5] Kim, Jinuk, et al. "GuidedQuant: Large Language Model Quantization via Exploiting End Loss Guidance." Forty-second International Conference on Machine Learning.

**Questions:**

1. Your evaluation exclusively uses RTN as a baseline. To even be considered for publication, your work must be situated within the current literature. Please provide direct, fair comparisons against both established standards (e.g., GPTQ, AWQ from 2023) and more recent state-of-the-art methods from 2024-2025 (e.g., DuQuant, OSTQuant, GPTAQ). Without this, your performance claims are unverifiable.
2. Please justify why the η scaling factor is presented as a novel contribution ("Enhanced Quantization by a Scaling Mechanism") rather than what it is: a basic hyperparameter for tuning the quantization range.
3. You motivate your work by the high cost of fine-tuning, then introduce a "layerwise-finetuning" step. Please provide a detailed computational cost analysis (GPU hours, memory, required calibration data size) for this step and explain how it is fundamentally more efficient than standard, full-model LoRA fine-tuning.
4. Please provide a clear ablation study that isolates the individual performance impact of: (a) η-scaling alone, (b) ALS optimization alone (without η-scaling), and (c) layerwise-finetuning on top of the other two. The current results are conflated and uninterpretable.

---

### Official Review · Reviewer_eGAU · 2025-10-31

**Soundness:** 2
**Presentation:** 1
**Contribution:** 1
**Rating:** 0
**Confidence:** 4

**Summary:**

This paper propose ALS-LoRA, a quantization method that uses iterative algorithm to initialise low-rank terms and fine-tuning to recover model performance.

**Strengths:**

The motivation and idea is straightforward

**Weaknesses:**

- Unclear concepts of QAT and PEFT: this paper mixes the concepts of w-only quantization and qLoRA. The former represent weights in low-precision, at inference time, the weights stored in memory are still low-precision. qLoRA only quantizes weights during fine-tuning, dequantize weights and fuse low-rank terms into weights after fine-tuning, so at inference time, qLoRA model is still un-quantized.
- The math/algorithms are ill-formed. For example,
  - Equation 1 in line 158, uses symbol `=` to denote the equivalence of objective, which is very rare. The author also applies l2-norm to matrix, but it should be Frobenius norm.
  - in algorithm 1, line 204, the update $W_q \leftarrow s, z$, which updates matrix using two scalar values, doesn't make sense.
- The objective is unclear. Both Alg 1 and Alg 2 seems to solve A and B, but it's unclear how these two algorithms serve the entire pipeline?
- Poor evaluation/experiment setup: For example
  -  In table 1, it seems author sweeps $\eta$ from 0 to 1 with a step size 0.1. May I ask why this hyper-param setup was adopted? How the quantile numbers in Table 1 are determined?
  - No other quantized PEFT baselines are involved in the paper. The author only compares ALS-LoRA against RTN and PTQ method Quarot and a uncommon number format (power of two)

**Questions:**

1. Is ALS-LoRA a quantization method that takes the form of $W_q X + AB X$ at inference time? I'm not very clear about the application scenario of this method
2. The purpose of the scaling factor $\eta$ is confusing. First, usually when people refer to round to nearest integer quantization in deep learning, they refer to the case where a floating point scale, if this is the case, the scalar $\eta$ (in line 5) is just absorbed into the floating point scale. Thus this should not change the model performance. Could the author elaborate this?
3. Why use ALS to solve the objective? If the objective in Eq (1) denotes minimizing the layer output error, there are already works giving its optimal solution, for example, [Caldera (NeurIPS2024)](https://arxiv.org/abs/2405.18886), [SVD-LLM (ICML2025)](https://arxiv.org/abs/2403.07378), [QERA (ICML2025)](https://openreview.net/forum?id=LB5cKhgOTu). **These paper uses SVD because that gives the optimal solution when the objective is to minimize the F-norm**. Does the iterative algorithm of ALS in Alg2 gives the same solution?
4. Why the number format of power of two is used as a baseline when ALS-LoRA uses integer quantization? This seems an unfair comparison. Also it will be convincing if comparison against SoTA quantization-aware training/fine-tuning baselines are included in the evaluation as the baselines in the evaluation sections are not strong enough.

To me the optimization objective, quite some notations, and most of the algorithms and evaluations do not make sense

---

### Official Review · Reviewer_7X15 · 2025-10-31

**Soundness:** 2
**Presentation:** 2
**Contribution:** 2
**Rating:** 4
**Confidence:** 2

**Summary:**

This paper proposes ALS-LoRA, a method that exploits ALS for low-rank matrix compensation without requiring full-model LoRA fine-tuning. This work has three contributions: a scaling mechanism that adjusts extreme values in RTN quantization to better handle outliers, using ALS instead of SVD to optimize low-rank compensation matrices, and layer-wise fine-tuning to improve compensation effects.

**Strengths:**

* Important problem
* Simple approach

**Weaknesses:**

* Novelty seems weak - the proposed method seems incremental
* Results seem a bit weak too, missing several baselines
* The writing needs improvement; abbreviations are repeatedly introduced without explanation.

**Questions:**

* There is no comparison with related work. How does this work compare to other related work?
* Could the authors provide results on computational savings?

---

### Meta-Review · Area_Chair_1dc7 · 2025-12-20

**Summary:**

**Paper summary.** This paper proposes ALS-LoRA, a low-rank compensation method for low-bit quantization. The method uses alternating least squares (ALS) to fit low-rank factors (using activation data), includes a scaling factor to adjust the quantization range, and optionally adds a small fine-tuning step. The reported results show gains over a naive round-to-nearest (RTN) quantizer, especially at 2–3 bits.

**What happened in the discussion.** The reviews were dominated by one major issue: the evaluation is not positioned against accepted baselines in the quantization literature. In particular, the strongest review calls out that comparing mainly against RTN is not acceptable in 2025+ because stronger PTQ baselines (e.g., GPTQ, AWQ, and more recent methods) are standard. Other reviews also pointed out conceptual confusion (mixing w-only quantization with qLoRA/QAT/PEFT settings) and issues in the math/algorithm presentation. There is no substantive author rebuttal in the forum export that addresses the baseline gap.

**My assessment as AC.** Even if the method idea is reasonable, the current paper does not provide the evidence needed to evaluate it: without fair comparisons to standard baselines, it is impossible to tell whether ALS-LoRA is competitive or just better than a strawman. In addition, unclear problem setup and presentation issues make the contribution hard to trust.

**Decision.** Reject. This is not a statement that the topic is unimportant. If the authors revise, the top priority is to rebuild the empirical section with fair baselines (GPTQ/AWQ and recent methods), clearly define the inference-time form and the training/quantization setting, and tighten the algorithm/maths. With a stronger evaluation and clearer setup, the work could be reconsidered at another venue.

**Reviewer Concerns:**

- **Missing strong baselines / literature omission (major)**: The paper does not compare against widely accepted quantization baselines; this undermines the empirical claims.
- **Conceptual confusion**: Reviewers noted conflation of w-only quantization with qLoRA/QAT/PEFT scenarios and unclear inference-time form.
- **Technical clarity and correctness**: Issues in equations/algorithm descriptions and writing quality.

**Reviewer Scores:**

- **Reviewer waPS (rating 2, confidence 5)**: Strongly negative; likely unchanged unless the evaluation is rebuilt with proper baselines.
- **Reviewer eGAU (confidence 4; rating 0)**: Negative on conceptual clarity and formulation; likely unchanged.
- **Reviewer 7X15 (rating 4, confidence 2)**: Mildly positive but also notes missing baselines and weak novelty; likely would move down with the same concerns emphasized by waPS.

---

### Decision · Program_Chairs · 2026-01-26

Reject